# MCW: A Generalizable Deepfake Detection Method for Few-Shot Learning

**DOI:** 10.3390/s23218763

**Published:** 2023-10-27

**Authors:** Lei Guan, Fan Liu, Ru Zhang, Jianyi Liu, Yifan Tang

**Affiliations:** 1Department of Electronic Engineering, Tsinghua University, Beijing 100190, China; guanl18@tsinghua.org.cn; 2Department of Cyberspace Security, Beijing University of Posts and Telecommunications, Beijing 100876, China; ysd@bupt.edu.cn (F.L.); liujy@bupt.edu.cn (J.L.); tyfcs@bupt.edu.cn (Y.T.)

**Keywords:** deepfake detection, meta-learning, few-shot, zero-shot

## Abstract

With the development of deepfake technology, deepfake detection has received widespread attention. Although some deepfake forensics techniques have been proposed, they are still very difficult to implement in real-world scenarios. This is due to the differences in different deepfake technologies and the compression or editing of videos during the propagation process. Considering the issue of sample imbalance with few-shot scenarios in deepfake detection, we propose a multi-feature channel domain-weighted framework based on meta-learning (MCW). In order to obtain outstanding detection performance of a cross-database, the proposed framework improves a meta-learning network in two ways: it enhances the model’s feature extraction ability for detecting targets by combining the RGB domain and frequency domain information of the image and enhances the model’s generalization ability for detecting targets by assigning meta weights to channels on the feature map. The proposed MCW framework solves the problems of poor detection performance and insufficient data compression resistance of the algorithm for samples generated by unknown algorithms. The experiment was set in a zero-shot scenario and few-shot scenario, simulating the deepfake detection environment in real situations. We selected nine detection algorithms as comparative algorithms. The experimental results show that the MCW framework outperforms other algorithms in cross-algorithm detection and cross-dataset detection. The MCW framework demonstrates its ability to generalize and resist compression with low-quality training images and across different generation algorithm scenarios, and it has better fine-tuning potential in few-shot learning scenarios.

## 1. Introduction

Some users of social networks utilize “deepfake” algorithms to swap a person’s face with another face to create fake videos. The usual forgery techniques include face swapping, full-face synthesis, attribute manipulation, and expression swapping [1]. Criminal individuals exploit these deepfakes to create pornographic videos, to propagate fake news, and to engage in the cyberextortion and financial fraud. To combat these cybercrimes caused by deepfakes, many forensics techniques are being discussed to detect manipulated faces in images and videos.

Researchers have explored various deepfake detection approaches [2,3,4]. Darius Afchar et al. introduced the MesoNet network based on mesoscopic characteristics [5]. Huy H. Nguyen et al. proposed CapsuleNet [6]. A Google team introduced the Xception architecture, replacing the Inception module with depthwise separable convolutions [7]. DY Zhang et al. designed a two-stream deepfake detection network SRTNet by combining the image spatial domain and residual domain [8]. These methods mainly focused on improving network structures, and detection performance relied on model tuning and dataset quality. Falko Matern et al. used the differences in biometric information between real and fake faces to identify manipulations [9]. Xin Yang et al. extracted features based on the disparity in facial landmarks to discern image authenticity [10]. Liu et al. proposed the utilization of global texture information in images [11]. These interpretable analysis modules based on biometric information significantly improved the detection performance. Xu Zhang et al. proposed a classifier based on frequency domain features [12]. Jiaming Li et al. developed a frequency-aware discriminative learning framework with an adaptive frequency feature generation module [13]. Liu et al. applied spatial phase shallow learning methods to extract the frequency components of the phase spectrum [14]. Afterwards, with the proposal of Transformer, many researchers used the attention mechanism to weight features of manipulated regions for detection deepfake [15,16,17,18,19].

A few papers considered detection in few-shot scenarios [20]. Lingzhi Li et al. proposed the face X-ray, which utilizes the boundaries generated during image fusion to detect inconsistent image features on both sides of the boundary to identify deepfake images [21]. Haonan Qiu et al. addressed the problem of few-shot deepfake detection by introducing a guided adversarial interpolation method. This method adversarially interpolates the visual artifacts of a few samples into the majority samples under the guidance of a teacher network, facilitating the learning of transferable distribution features between different domains and enhancing the model’s detection capability for few-shot data [22]. YK Lin et al. proposed two methods to discuss few-shot detection: a Few-Shot Training GAN for Face Forgery Classification and Segmentation Based on the Fine-Tune Approach, which increases detection accuracy with the same training and testing dataset [23], and a forged image detector using meta-learning techniques to detect forged regions using a smaller number of training samples, which increases detection accuracy with zero-shot training [24].

Ke Sun et al. proposed the LTW algorithm based on the meta-weight learning algorithm, which simulates cross-domain processes during the training phase to assign different weights to each training sample, enabling the model to learn highly generalizable features [25]. The LTW first applied meta-learning methods to the field of false face detection, but the input is simple and only uses RGB domain images, without paying attention to the frequency domain features of the images. Moreover, the utilization of meta-knowledge is too rough, with fine-grained data only reaching the sample level, resulting in poor guidance effectiveness of meta-networks. Davide Cozzolino et al. proposed the Forensictransfer (FT) algorithm based on autoencoders [26]. By minimizing reconstruction errors for training, an encoder is obtained that can constrain true and false face images in the embedding space, where the embedded features preserve all the expected information. However, research lacks a more detailed utilization of images and has not expanded the visible range of models trained using source domain data. Shivangi Aneja et al. proposed the Deep Distribution Transfer Method (DDT), which utilizes domain adaptation-based transfer learning methods [27]. The DDT associates some target test samples with one of the previously trained patterns and migrates the model to a new domain. And some data augmentation was carried out during the fine-tuning stage. The disadvantage is that the learning feature of transfer learning is not effective, and the data augmentation during the fine-tuning stage does not produce substantial data augmentation [28].

In this paper, we propose a multi-feature channel domain-weighted framework based on meta-learning (MCW). By improving the drawbacks of the previous algorithm, the MCW enables the capture of cross-domain invariant features while avoiding the learning of unnecessary semantic and background features. We employ the meta-learning method to learn meta-weight knowledge based on tasks from high-frequency and RGB image information. At the same time, features are weighted to facilitate fine-grained feature mining for generalization purposes. In all, we improve existing rough data preprocessing methods and facial data augmentation methods, constructing a feature-level data augmentation strategy.

The main contributions of this paper are as follows:We propose a multi-feature channel domain-weighted framework based on meta-learning. Two improvement methods are proposed: combining the RGB domain and frequency domain information of the image to enhance the feature extraction ability of the model in detecting targets, and assigning meta-weights to channels on the feature map to enhance the model’s generalization ability for detecting targets.We improve the existing rough data preprocessing methods and facial data augmentation methods, constructing a feature-level data augmentation strategy. The improvement of this strategy is helpful for deepfake recognition.Through comparative experiments, we prove that the MCW has advantages in deepfake detection in a few-shot scenario. The proposed MCW can serve as a reference for future research in real situations.

In the rest of the paper, the key components of the proposed method are detailed in Section 2. Section 3 presents the experimental procedures and analysis of the results. Finally, in Section 4, we discuss and provide conclusions with potential future works.

## 2. Framework

This framework utilizes two networks: the basic network denoted as f(θ), and the meta-weight network denoted as m(ω), as shown in Figure 1. During each training epoch, MCW randomly divides the dataset D into a support set and a query set. The basic network f(θ) first updates the parameters using the support set, and then the meta-weight network updates the parameters using the query set.

The basic network of the MCW is EfficientNet-b4. The weight meta-network comprises a global pooling layer and two fully connected layers. The global pooling layer compresses each channel while retaining the number of channels. The subsequent fully connected layers would learn enough nonlinear relationships among the channels. Additionally, dimension reduction is performed before increasing the dimensions between the two fully connected layers, thereby reducing the network parameters.

### 2.1. Data Preprocessing

In this stage, each video is evenly divided into 10 frames, and each frame is cropped using the MTCNN algorithm. Dataset D is randomly divided into two halves: $D_{meta-train}$ and $D_{meta-test}$, respectively, as the support set and the query set.

### 2.2. Meta-Training and Meta-Testing

In the meta-training stage, θ represents the parameters of the basic network, and ω represents the parameters of the meta-network. K training data are extracted from $D_{meta-train}$, as $X_{s}=\{x_{i},y_{i}{\}}_{i=1}^{K}$. The loss function is denoted as L. The weighted loss is denoted as T(θ,ω). Use this weighted loss to update the parameters, which are pseudo parameters θ′. The formula is as follows:(1)$$T\left(\theta ,\omega \right)=\frac{1}{K}\sum_{i=1}^{K}L((x_{i},y_{i});\theta ,\omega ),$$
(2)$$\theta^{\prime }=\theta -\alpha \nabla_{\theta }T\left(\theta ,\omega \right).$$

In the meta-testing stage, we retrieve the data $X_{s}^{\prime }=\{x_{i}^{\prime },y_{i}^{\prime }{\}}_{i=1}^{K}$ from the corresponding query sets $D_{meta-test}$. Using the pseudo parameters θ’ obtained in the previous step, we calculate the loss in the meta-testing stage, denoted as $M(\theta^{\prime },w)$, according to the following formula:(3)$$M(\theta^{\prime },\omega )=\frac{1}{K}\sum_{i=1}^{K}L((x_{i}^{\prime },y_{i}^{\prime });\theta^{\prime },\omega ).$$

After obtaining the losses for both stages, the parameter update goal is defined as follows:(4)$$argminT(\theta ,\omega )+\beta M(\theta^{\prime },\omega ).$$

Here, β is a user-defined parameter. Consequently, the parameters of the basic network are fully updated as:(5)$$\theta^{*}=\theta -\alpha \nabla_{\theta }T(\theta ,\omega )+\beta (\theta^{\prime }-\gamma \nabla_{\theta^{\prime }}M(\theta^{\prime },\omega ).$$

The meta-network’s parameters are updated as:(6)$$w^{*}=w-\phi \nabla_{w}M(\theta^{\prime },w).$$

### 2.3. Loss Function

To enhance the generalization of the deepfake detection, MCW incorporates two loss functions.

#### 2.3.1. Binary Center Loss

Because the softmax function sometimes results in a larger intra-class distance than the inter-class distance, MCW uses the center loss to minimize the distance between feature points and center points of the category. The center loss function is represented by Formula (7), where $c_{y_{i}}$ denotes the center point of the $y_{ith}$ category, $x_{i}$ represents the feature in front of the fully connected layer, and m is the mini-batch size.
(7)$$L_{C}=\frac{1}{2}\sum_{i=1}^{m}||x_{i}-c_{y_{i}}|{|}_{2}^{2}.$$

However, deepfake detection involves binary classification. While maintaining a center point for the true class to enhance compactness is reasonable, doing the same for the false class is not. The training set consists of true data and false data generated from various forgery methods, resulting in different probability distributions. Optimizing the inter-sample distance of all false data as one category might lead to distribution overlap. To address this problem, an improved binary center loss function, denoted as $L_{BC}$, is proposed in this subsection.
(8)$$L_{BC}=\frac{1}{2}\sum_{i=1}^{N_{real}}||f(x_{i}^{real})-C_{real}|{|}_{2}^{2}-\frac{1}{2}\sum_{i=1}^{N_{fake}}||f(x_{i}^{fake})-C_{real}|{|}_{2}^{2},$$
(9)$$C_{real}=\frac{1}{N_{real}}\sum_{i=1}^{N_{real}}f(x_{i}^{real}),$$
where $C_{real}$ denotes the center point of the true class and $N_{real}$ represents the number of true samples in a mini-batch. Compared to the original center loss function, the binary center loss function separately calculates the center distance of true samples and false samples. The objective is to minimize the distance of true samples and maximize the distance of false samples to the true center point. No center point of the false samples is defined.

#### 2.3.2. Binary Cross-Entropy Loss

The binary cross-entropy loss is a common loss function for binary classification problems. It quantifies the difference between the predicted distribution and the real distribution. The objective is to minimize the dissimilarity between the predicted labels and the real labels. When the predicted labels exactly match the real labels, the loss function reaches the minimum value of 0. Binary cross-entropy loss is denoted as $L_{BCE}$.

The total loss of the MCW is denoted by L, as shown in following equation:(10)$$L=L_{BCE}+\lambda L_{BC}.$$

## 3. Experiments and Analysis

### 3.1. Dataset

To validate the generalization and robustness of the proposed algorithm, we conducted experiments using two datasets: FaceForensics++ (FF++) and Google DFD [29]. The FF++ dataset contains fake face data generated by four different algorithms, making it suitable for cross-domain generalization testing. The DFDC, Celeb-DF, and Google DFD datasets were used as supplementary data to broaden the scope of the experiments [30,31].

For the meta-training and meta-testing, we utilized the FF++ dataset. We selected 720 videos for the training set, 140 videos for the validation set, and an additional 140 videos for the test set. Moreover, we randomly chose 300 real-face videos and their corresponding fake-face videos from the DFDC dataset, along with 500 test videos from the Celeb-DF dataset. From the DFD dataset, we handpicked 28 real-face videos from specific locations and paired them with corresponding deepfake fake-face videos.

To simulate scenarios with limited test video samples or a small number of images, as commonly encountered in forensic settings, we extracted only 10 evenly spaced frames from each video. We employed the MTCNN algorithm to detect and segment the face regions, resizing them to 224*224 pixels. Specifically, we used the C23 and C40 compression rate data from the FF++ dataset, excluding the raw data. These settings allowed us to assess the generalization performance of each model under conditions with limited data and poor data quality.

### 3.2. Settings

The DeepFake detection framework uses a learning rate α of 0.001 for the meta-training stage of the classifier, and a learning rate γ of 0.001 for the meta-testing stage. A fixed step decay (StepLR) strategy is employed for learning rate decay, with a step size of 5 and a gamma value of 0.1. The Adam optimizer was chosen, and the overall classification network parameter update is controlled by a hyperparameter β set to 1. The training process uses a batch size of 16 and continues until stability is reached, typically around 30 epochs.

To comprehensively assess the performance of the proposed MCW framework in terms of cross-domain generalization, it includes two sets of comparative experiments as baselines. These experiments evaluate the performance of the proposed method in cross-forgery-method and cross-dataset scenarios.

To evaluate the performance in cross-forgery-method scenarios, six DeepFake detection algorithms specifically designed for addressing cross-forgery-method generalization were selected for comparison:Xception. A model trained directly on the entire training set, which serves as a baseline standard. Reference [6].Face X-ray. Reference [21].LTW. An algorithm based on the meta-weight learning algorithm. Reference [25].SupCon. The Supervised Contrastive Learning method for generalization and interpretable deepfake detection. Reference [32].MLDG. The Meta-Learning Domain Generalization method. Reference [33].MLA. A Meta-Learning Approach for Few-Shot Face Forgery Segmentation and Classification. Reference [23].

To evaluate performance in cross-dataset scenarios, four additional state-of-the-art domain adaptation and few-shot learning methods were included: Prototypical Networks and Relation Networks for few-shot learning [34,35]. Also, we choose the Forensictransfer (FT) algorithm and the Deep Distribution Transfer (DDT) method for weakly supervised domain adaptation in deepfake detection [26,27].

### 3.3. Results and Analysis

#### 3.3.1. Zero-Shot Scenario: Cross-Forgery-Method Evaluation on FF++ Dataset

The zero-shot scenario involves training the classifier on the source domain and directly testing it on the target domain. In this subsection, we present the results of cross-algorithm detection experiments on four subsets of the FF++ dataset: DeepFake (DF), Face2Face (F2F), FaceSwap (FS), and NeuralTextures (NT). To simulate DeepFake videos processed at different compression rates in real-world scenarios, we selected two compression rate data: C23 and C40. Three subsets were used as the training set, and the remaining subset was used as the test set. For example, “Others-DF” indicates that the training set consists of F2F, FS, and NT, and the test set is DF. The text before “-” symbol denotes the training set, and the text after denotes the test set.

We first present the comparison results of high compression rate scenario (C40) in Table 1. The bold parts in the table represent the best results. As noted from Table 1, in the case of high compression rates, the proposed MCW algorithm outperforms the baseline methods on most of the benchmarks. Although the LTW method achieved a higher accuracy (ACC) than the MCW in the DF and F2F test sets, MCW achieved the highest area under the curve (AUC) and ACC compared to the other benchmarks. This indicates that the MCW algorithm, which incorporates frequency domain information, exhibits a stronger advantage and robustness in dealing with high compression rate scenarios compared to similar algorithms. Additionally, the training approach of the meta-learning framework in the MCW enables stronger cross-algorithm detection capability when facing data generated by different algorithms. The Face X-ray method, which demonstrated good generalization ability in its original paper, performed poorly in these experimental settings, with an AUC value 10% lower than that of the MCW algorithm. Observations on the training data reveal that, in this experiment, some facial data exhibited missing regions around the face bounding box due to the fixed pixel size, making accurate mask extraction challenging. Moreover, only 10 frames were used for each video, which limits the availability of facial data for the Face X-ray algorithm. Furthermore, the lower image quality may affect the performance of the Face X-ray method, which indicates its weaker resistance to compression. Lastly, as the source code of the Face X-ray algorithm is not publicly available, the existing implementation may not achieve the performance described by the authors.

We further present the comparison results of the low compression rate scenario (C23) in Table 2. As noted from Table 2, in the low compression rate scenarios, the MCW algorithm consistently achieved a high performance, exhibiting the best performance in most benchmarks with the highest AUC value reaching 0.929. However, when DF was used as the test set, the ACC value of the MCW was 0.9% lower than that of the Face X-ray algorithm. Similarly, when FS was used as the test set, the ACC and AUC of MCW are 1% and 2.1% lower than those of the Face X-ray algorithm, respectively. Nonetheless, MCW consistently maintained the best overall performance across the remaining benchmarks. This indicates that in low compression rate scenarios with high image quality, the MCW’s lead over other algorithms becomes smaller. However, overall, the MCW still demonstrates superior cross-domain generalization ability. The performance of the Face X-ray algorithm under the low compression rate (C23) was significantly better than its performance under the C40 compression rate. It outperformed the MCW in the DF and FS test sets. This suggests that the Face X-ray algorithm has specific requirements for image quality, and higher-quality images contribute to obtaining a higher-quality auxiliary dataset.

#### 3.3.2. Few-Shot Scenario: Cross-Forgery-Method Evaluation Using the FF++ Dataset

To evaluate the adaptability of the MCW algorithm using the meta-learning framework, a fine-tuning approach was employed to test its performance and compare it with several other algorithms. Fine-tuning is generally applied to refine models for new tasks. In this subsection, we provide experimental results to evaluate the effectiveness of the fine-tuning model using different data distributions.

Specifically, the first 5 and 10 videos from the current test data were used for fine-tuning. The fine-tuning results were evaluated using 50 images. In the MCW framework, all networks’ parameters, except for the last fully connected layer, were frozen. The initial learning rate (lr) was set to 0.0001.

As noted from Table 3, fine-tuning the model with a small number of unknown domain samples from the test set effectively improved the model’s accuracy (ACC) and area under the curve (AUC) metrics in the unknown domain. This process contributes to enhancing the model’s generalization ability, enabling better performance in new domains. Among the three models incorporating the ideas of meta-learning, more significant performance improvements were observed during the fine-tuning process. This finding suggests that meta-learning enables the models to learn more promising parameters and capture more domain-invariant features compared to conventional training methods.

#### 3.3.3. Zero-Shot Cross-Dataset Generalization Evaluation

In this subsection, we present the results of the cross-dataset scenario using the FF++, DFDC, Celeb-DF, and DFD datasets. The FF++ dataset was utilized with a C23 compression rate. Due to the specific nature of the MCW algorithm, it was necessary to simulate distribution shift using data with different distributions. Therefore, the training set consisted of four subsets of the FF++ dataset (DF, F2F, FS, and NT), while the test set included the DFDC, Celeb-DF, and DFD datasets.

Under the high compression rate C23 scenario, as noted from Table 4, the performance of the MCW algorithm remained stable, maintaining a high level of performance. It achieved the highest performance in the DFDC dataset test, with an AUC value 1.4% higher than that of Face X-ray. However, in the FF++-Celeb-DF dataset test, the AUC value of the MCW was 2.5% lower than that of Face X-ray. Overall, in terms of cross-dataset performance, the MCW algorithm is comparable to Face X-ray, without significant advantages. This suggests that the MCW algorithm performs better in terms of high compression rates and cross-generation algorithms, while still exhibiting good performance in cross-dataset scenarios. Face X-ray demonstrates good performance when trained on high-quality images. Since this algorithm utilizes real face data for training, it imposes higher requirements on image quality.

#### 3.3.4. Few-Shot Cross-Dataset Generalization Evaluation

In this subsection, we present the cross-dataset generalization performance of the MCW algorithm in the few-shot scenario. We conducted a comparative test using 50 images for fine-tuning, as presented in Table 5. The MCW algorithm proposed in this study demonstrated superior performance compared to the comparison methods in almost all aspects.

#### 3.3.5. Ablation Study

To comprehensively evaluate the contributions of different modules in the MCW algorithm framework, we conducted ablation experiments. We partitioned the original algorithm framework into two variants: MCW/MetaModule, which employed the regular training approach with unchanged parameters, and MCW/FeaFusion, which excluded the FeaFusion module responsible for feature fusion. Subsequently, we compared the results obtained from these variants, assessing their generalization performance across the four subsets of the FF++ dataset at a data compression rate of C23.

As noted from Table 6, the complete MCW algorithm consistently achieved the highest performance across all data splits. When the MetaModule and FeaFusion modules were separated, the performance decreased, with an average AUC reduction of 1.25% and 1.17%. This outcome demonstrates the effectiveness of fusing RGB and frequency domain features in enhancing the model’s cross-domain generalization performance and robustness. Moreover, the channel-wise weight assignment in meta-learning confers higher weights to channels housing cross-domain invariant features, thereby enhancing the model’s generalization capability.

## 4. Conclusions and Future Work

This paper presented the MCW framework, a channel-domain weighted meta-learning approach based on multiple features. The framework integrates RGB domain features with high-frequency and low-frequency domain features and uses meta-learning for training, wherein the meta-network calculates channel weights. The experimental results show that the proposed approach significantly enhances the model’s performance in scenarios with limited data and compression, thereby improving its generalization ability and robustness.

The experiment described in this article used C23 and C40 compression rate data from the FF++dataset, and only 10 frames of the images were used for each video. The experimental setup simulates scenarios where data are compressed and training data are missing. The experimental environment is close to a real scene. Compared with the other nine algorithms, the proposed MCW framework outperforms other algorithms in cross-algorithm detection and cross-dataset detection. Therefore, we consider that the proposed MCW can effectively complete deepfake detection tasks in real situations.

However, under the normal use of dataset training, the two methods proposed in this article still have some optimization space, and their performance is comparable or inferior to the best methods in the field of deepfake detection generalization that are currently being studied, with a certain performance gap. The method in this article aims to consider the problem of false face detection from the perspective of small samples and domain generalization, and does not pursue the improvement of detection speed. The use of meta-learning increases the cost of model training and learning, which is the drawback of this method.

Our future research will focus on the following aspects:

Optimizing meta-learning methods to calculate costs. The training of meta-learning usually requires a large amount of computational resources. We will continue to try lightweight models and use incremental learning methods to reduce training costs in the future.Enhancing the utilization of temporal information in the model. The MCW is used for training and detecting images extracted from video segmentation. Deepmake videos have time information. Our method lacks detection of image motion and sound features. Adding multimodal feature detection can improve the detection accuracy of fake face videos.

## Figures and Tables

**Figure 1 sensors-23-08763-f001:**
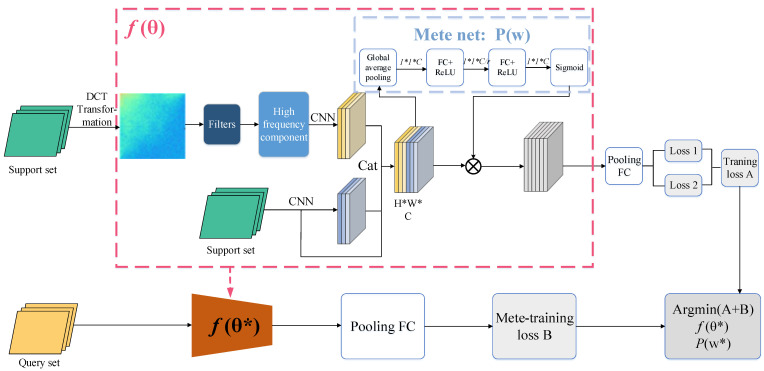
Multi-feature channel domain-weighted meta-learning framework. The basic network denoted as f(θ), and the meta-weight network denoted as m(ω).

**Table 1 sensors-23-08763-t001:** Test results on FF++ dataset (C40 compression rate).

Method	Others-DF		Others-F2F		Others-FS		Others-NT	
	ACC	AUC	ACC	AUC	ACC	AUC	ACC	AUC
Xception	0.663	0.721	0.602	0.621	0.577	0.609	0.563	0.601
MLDG	0.671	0.730	0.581	0.617	0.581	0.617	0.569	0.607
SupCon	0.67	0.735	0.623	0.601	0.582	0.624	0.571	0.611
LTW	**0.691**	0.756	**0.657**	0.724	0.625	0.681	0.585	0.608
Face X-ray	0.661	0.712	0.611	0.623	0.534	0.598	0.577	0.621
MCW (ours)	0.690	**0.771**	0.654	**0.731**	**0.641**	**0.702**	**0.591**	**0.625**

**Table 2 sensors-23-08763-t002:** Test results on FF++ dataset (C23 compression rate).

Method	Others-DF		Others-F2F		Others-FS		Others-NT	
	ACC	AUC	ACC	AUC	ACC	AUC	ACC	AUC
Xception	0.827	0.898	0.647	0.782	0.497	0.597	0.556	0.761
MLDG	0.842	0.918	0.634	0.771	0.527	0.609	0.621	0.78
MLA	0.666	-	0.647	-	0.482	-	0.557	-
SupCon	0.839	0.901	0.647	0.793	0.5	0.603	0.556	0.767
LTW	0.856	0.927	0.656	0.802	0.549	0.64	0.653	0.773
Face X-ray	**0.871**	0.912	0.631	0.779	**0.572**	**0.672**	0.656	0.781
MCW (ours)	0.862	**0.929**	**0.657**	**0.811**	0.562	0.651	**0.674**	**0.788**

**Table 3 sensors-23-08763-t003:** Fine-tuning results on FF++ dataset (C23 compression rate)—50 images.

Method	Others-DF		Others-F2F		Others-FS		Others-NT	
	ACC	AUC	ACC	AUC	ACC	AUC	ACC	AUC
Xception	0.844	0.902	0.661	0.813	0.538	0.623	0.594	0.813
MLDG	0.876	0.931	0.667	0.821	0.556	0.639	0.673	0.832
LTW	**0.884**	**0.937**	0.687	0.841	0.601	0.691	0.704	0.845
MCW (ours)	0.881	0.935	**0.701**	**0.855**	**0.625**	**0.711**	**0.711**	**0.849**

**Table 4 sensors-23-08763-t004:** Cross-dataset test results.

Method	FF++-DFDC		FF++-Celeb-DF		FF++-DFD	
	ACC	AUC	ACC	AUC	ACC	AUC
Xception	0.622	0.685	0.632	0.655	-	-
MLDG	0.607	0.682	0.595	0.609	-	-
SupCon	0.603	0.674	0.621	0.633	-	-
Face X-ray	0.633	0.689	0.671	**0.692**	-	-
Prototypical Nets	-	-	0.58	-	0.713	-
Relation Nets	-	-	0.632	-	0.728	-
FT	-	-	0.478	-	0.678	-
DDT	-	-	**0.688**	-	0.812	-
MCW (ours)	**0.644**	**0.703**	0.642	0.667	**0.817**	**0.823**

**Table 5 sensors-23-08763-t005:** Fine-tuning results with 50 images in cross-dataset testing.

Method	FF++-DFDC		FF++-Celeb-DF		FF++-DFD	
	ACC	AUC	ACC	AUC	ACC	AUC
Xception	0.658	0.717	0.671	0.722	-	-
MLDG	0.657	0.702	0.655	0.698	-	-
SupCon	0.644	0.694	0.662	0.716	-	-
LTW	0.681	0.743	0.699	0.757	-	-
Prototypical Nets	-	-	0.65	-	-	-
Relation Nets	-	-	0.687	-	-	-
FT	-	-	0.69	-	-	-
DDT	-	-	**0.739**	-	-	-
MCW (ours)	**0.695**	**0.758**	0.712	**0.768**	**0.849**	**0.876**

**Table 6 sensors-23-08763-t006:** Ablation experiment results using FF++ dataset (C23 compression rate).

Method	Others-DF		Others-F2F		Others-FS		Others-NT	
	ACC	AUC	ACC	AUC	ACC	AUC	ACC	AUC
Xception	0.827	0.898	0.647	0.782	0.497	0.597	0.556	0.761
MCW/MetaModule	0.855	0.921	0.646	0.79	0.524	0.635	0.649	0.783
MCW/FeaFusion	0.855	0.928	0.651	0.803	0.521	0.622	0.644	0.779
MCW	**0.862**	**0.929**	**0.657**	**0.811**	**0.562**	**0.651**	**0.674**	**0.788**

## Data Availability

Not applicable.

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
