# Peer review of "MCW: A Generalizable Deepfake Detection Method for Few-Shot Learning"

_sensors, 2023, doi:10.3390/s23218763_

Round 1
Reviewer 1 Report
The topic is suitable for the journal. The author should explain more about the main contribution of the manuscript. However, the following significant Major corrections are necessary to improve the scientific level of the manuscript.
- Please improve the abstract section; it needs to deliver more information about the manuscript to readers. It should have a conclusion section. The method's name and acronym (MCW) need to be mentioned.
- Please explain more clearly the main contribution. The authors need to highlight the novelty of the work presented.
- Why do authors think they are appropriate for such an application? What is their main advantage?
- Please explain clearly the limitations and future work.
- The complexity of the proposed model and the model parameter's uncertainty are not mentioned.
- A reasonable justification should be made about why such algorithms are used. Why do authors think they are appropriate for such an application? What is their main advantage over other methods? The literature review is not extensive. The literature review is long but needs more content with essential subjects. Also, not to mention recent up-to-date research work. Much related recent research is available online.
Reviewer 2 Report
The paper title sounds good but it has some major issues regarding the way it is presented.
The abstract needs to be improved as it does not contain enough information about the achieved results of the proposed approach in comparison to existing approaches.
There are unclear abbreviations such as MCW. Please check for the rest of the paper.
The work misses the literature review. You need to find the gaps in the existing solution to propose your own solution. Also, you need to discuss how the existing work has tackled the problem.
Figure 1 needs to be referred to in the context of your work.
Figure 1 is outside the margins of the paper.
There is a data set and tables of the results but where are the actual plots/ results you achieved? You need to show those plots so we can compare what you have achieved and what others achieved. We need to see the evaluation of your work.
The conclusion needs to be rewritten to reflect what you have discussed and achieved in this work.
Good but few sections need to be rewritten
Reviewer 3 Report
The paper is like a report for bachelor degree not as full research paper.
The article has many weaknesses
1. The abstract is very weak and simple
2. There is no discussion about the literature review about the topic
Moderate editing of English language required
Round 2
Reviewer 1 Report
The revision of this manuscript is satisfied.
The revision of this manuscript is satisfied.
Reviewer 2 Report
I am satisfied with how the reviewed version looks like.
Reviewer 3 Report
No further comments